DATA RELEASE

# A phased, chromosome-scale genome of 'Honeycrisp' apple (*Malus domestica*)

Awais Khan[1,*], Sarah B. Carey[2,3], Alicia Serrano[1], Huiting Zhang[4,5], Heidi Hargarten[4], Haley Hale[2,3], Alex Harkess[2,3] and Loren Honaas[4,*]

1 Plant Pathology and Plant-Microbe Biology Section, Cornell University, Geneva, NY 14456, USA
2 Department of Crop, Soil, and Environmental Sciences, Auburn University, Auburn, AL 36849, USA
3 HudsonAlpha Institute for Biotechnology, Huntsville, AL 35806, USA
4 USDA ARS Tree Fruit Research Lab, Wenatchee, WA 98801, USA
5 Department of Horticulture, Washington State University, Pullman, WA, USA

## ABSTRACT

The apple cultivar 'Honeycrisp' has superior fruit quality traits, cold hardiness, and disease resistance, making it a popular breeding parent. However, it suffers from several physiological disorders, production, and postharvest issues. Despite several available apple genome sequences, understanding of the genetic mechanisms underlying cultivar-specific traits remains lacking. Here, we present a highly contiguous, fully phased, chromosome-level genome of 'Honeycrisp' apples, using PacBio HiFi, Omni-C, and Illumina sequencing platforms, with two assembled haplomes of 674 Mbp and 660 Mbp, and contig N50 values of 32.8 Mbp and 31.6 Mbp, respectively. Overall, 47,563 and 48,655 protein-coding genes were annotated from each haplome, capturing 96.8–97.4% complete BUSCOs in the eudicot database. Gene family analysis reveals most 'Honeycrisp' genes are assigned into orthogroups shared with other genomes, with 121 'Honeycrisp'-specific orthogroups. This resource is valuable for understanding the genetic basis of important traits in apples and related Rosaceae species to enhance breeding efforts.

**Subjects** Genetics and Genomics, Agriculture, Plant Genetics

**Submitted:** 14 July 2022

\* Corresponding authors. E-mail: awais.khan@cornell.edu; loren.honaas@usda.gov

Preprint submitted at https://doi.org/10.1101/2022.08.24.505160

## BACKGROUND

Apples are the most consumed fruit in the United States [1]. The annual estimated total value of the US apple industry is $23 billion, with five cultivars alone accounting for two-thirds of production (in order of proportion): 'Gala', 'Red Delicious', 'Honeycrisp', 'Granny Smith', and 'Fuji' [2]. Of these, 'Honeycrisp' is by far the most valuable: it has roughly twice the value per pound of the next most valuable cultivar, 'Fuji' [3]. 'Honeycrisp' is appreciated by consumers, and therefore by the US apple industry, for its superior flavor and crisp, juicy texture. Importantly, properly stored 'Honeycrisp' fruit can be well-preserved for several months [4, 5]. Additionally, this cultivar shows high levels of cold hardiness [6] and resistance to apple scab, the most economically important fungal disease of apples worldwide [7]. 'Honeycrisp' was bred at the University of Minnesota in the 1960s, where the aim was to obtain cold hardy cultivars with high-quality fruit; it was released in 1991 [8] (Figure 1A). Recent genome-wide analysis (following the resolution of the 'Honeycrisp' pedigree [9, 10]) showed that the genetic background of 'Honeycrisp' is distinct from other important apple cultivars in the USA. This is highlighted by the success of 'Honeycrisp' as a source of interesting genetic diversity in apple breeding programs

**Figure 1. Physiology, and physiological disorders, of 'Honeycrisp' apple.**
(A) Healthy 'Honeycrisp' apples. (B) 'Honeycrisp' apples with symptoms of zonal leaf chlorosis. 'Honeycrisp' apples with symptoms of the fungal diseases (C) bitter rot pathogen complex (*Colletotrichum gloeosporiodes* and *C. acutatum*) and (D) black rot pathogen (*Botryosphaeria obtuse*). 'Honeycrisp' apples with the postharvest storage disorders (E) bitter pit, (F) soft scald, and (G) soggy breakdown.

worldwide to enhance texture, storability, and improved disease resistance [5, 7, 9, 11, 12]. In fact, nine new cultivars derived from 'Honeycrisp' are already on the market.

Although critical for sustainable apple production, disease resistance has historically been less important because the market has been dominated by modern cultivars bred primarily for fruit quality and intensive conventional production systems [13]. Most apple cultivars grown commercially in the USA are susceptible to fungal diseases such as apple scab. In temperate and humid regions around the world, frequent applications of fungicides are necessary, contributing significantly to production costs, and to negative human health and environmental impacts [14]. 'Honeycrisp' is resistant to apple scab and, importantly, the ability of the fruits of this cultivar to retain crispness and firmness during storage is one of its most outstanding traits [15]. However, other 'Honeycrisp' production issues present challenges for apple growers (Figure 1E–G). 'Honeycrisp' needs a carefully designed nutrient management program during the growing season for optimal production and fruit quality, especially to limit the occurrence of the physiological disorder bitter pit [5]. 'Honeycrisp' trees also have greater tendency to develop zonal leaf chlorosis, which reduces photosynthetic capacity [16]. However, in the Pacific Northwest (PNW), where most 'Honeycrisp' apples in the USA are grown [17] because of the low disease pressure in this region, postharvest issues during long-term storage pose substantial challenges to producers.

The total cullage of 'Honeycrisp' fruit is probably among the highest of apple cultivars. This is because of its susceptibility to various postharvest physiological disorders with poorly understood and complex etiologies. Such etiologies include bitter pit, soft scald, soggy breakdown, and $CO_2$ injury [18–21]. Postharvest technologies have been developed and deployed to mitigate these disorders [22–24]. However, factors affecting the efficacy of postharvest treatments include preharvest orchard management and at-harvest fruit maturity – key in the maintenance of postharvest apple fruit quality. Growers must balance the acquisition of certain fruit quality characteristics (e.g., size, color, flesh texture, and sugar content), while attempting to minimize risk for maturity-linked losses in quality that may occur in the supply chain [25]. This balancing act for maximizing at-harvest fruit quality and long-term cold storage potential in controlled atmospheres is especially difficult for 'Honeycrisp'.

## CONTEXT

To maximize both our understanding of genetic mechanisms driving important 'Honeycrisp' traits, and to assist tree fruit breeders, high quality genomes are required [26]. Indeed, in the last decade since 'Golden Delicious' was sequenced [27], many genes and quantitative trait loci (QTL) linked to fruit disease resistance, quality traits, and abiotic stress tolerance in apples have been identified [7, 28, 29]. Recent high-quality genomes of 'Gala', the double haploid 'Golden Delicious', and the triploid 'Hanfu' provide genomic resources for apple genetics and breeding [27, 30, 31]. These studies have identified targeted genomic regions for the development of diagnostic molecular markers to breed disease-resistant apple cultivars with good fruit quality [32]. However, traditional apple breeding is still resource-intensive and a time-consuming process [11, 29, 32]. Substantial gaps remain in our knowledge of the genetic mechanisms involved in many important apple traits. Here, we report a phased, chromosome-level genome assembly of the 'Honeycrisp' apple cultivar generated from Pacific Biosciences (PacBio) HiFi and Dovetail Omni-C technologies, plus a high-quality annotation, thus providing one of the most contiguous and complete genome resources available for apples to date.

## METHODS

### PacBio HiFi sequencing

Cuttings of dormant wood were collected from 'Honeycrisp' trees growing in the experimental orchard at Cornell AgriTech (Geneva, NY, USA). The cuttings were placed in water in the greenhouse until leaves began to emerge from buds, and thereafter placed in the dark for 2 days. Young, dark-adapted leaves were collected and shipped on dry ice to the DNA Sequencing and Genotyping Center at the University of Delaware (DL, USA) for DNA extraction and Single Molecule Real Time (SMRT) Pacific BioSciences (PacBio) sequencing.

High-molecular-weight (HMW) genomic DNA was extracted using a DNeasy Plant Mini Kit (Qiagen) according to the manufacturer's protocol. HMW genomic DNA was sheared to 15 kilobase pair (Kbp) fragments, and the HiFi library was prepared using SMRTbell Express Template Prep Kit 2.0 and the DNA/Polymerase Binding Kit 2.0 (Pacific Biosciences) according to the manufacturer's protocol. The sequencing library was size-selected using Sage Blue Pippin (Sage Sciences) to select fragment sizes of >10 Kbp to ensure removal of smaller fragments and adapter dimers. The library was sequenced on a PacBio Sequel II instrument in CCS/HiFi mode with two SMRT cells with 2 hours of pre-extension and



30-hour movie times. Read length distribution and quality of all HiFi reads was assessed using Pauvre v0.1923 [33].

To scaffold the genome using chromatin conformation sequencing, 1 g of flash-frozen young leaf material was harvested from 'Honeycrisp' trees at the Washington State University (WSU) Sunrise Research Orchard near Rock Island, WA USA and shipped to the HudsonAlpha Institute for Biotechnology in Huntsville, AL USA. The sequencing library was prepared using the Dovetail Genomics Omni-C kit and was sequenced on an Illumina NovaSeq 6000 with PE150 reads. A subset of 1 million read pairs was used as input for Phase Genomics hic_qc to validate the overall quality of the library [34].

## Phased haplome assembly and scaffolding

The expected genome size, heterozygosity, and percent of repeats was assessed by generating 21-mer sequences from the raw HiFi data with Jellyfish v2.3.0 (RRID:SCR_005491) [35] and GenomeScope 2.0 (RRID:SCR_017014) [36, 37]. HiFi reads were assembled into contigs using hifiasm v0.16.1 (RRID:SCR_021069) [38, 39], with the Hi-C integration mode that incorporated Dovetail Omni-C reads for phasing. Both haplomes of the assembly were scaffolded into chromosomes using the Juicer pipeline v1.6 (RRID:SCR_017226) [40], where the Omni-C reads were mapped separately to both hifiasm haplomes [39, 41] with the parameter "-s none". The Omni-C data was subset to ~100× coverage and the 3D-DNA v201008 scaffolding pipeline [42] was run with options "–editor-saturation-centile 10 –editor-coarse-resolution 100000 –editor-coarse-region 400000 –editor-repeat-coverage 50". Contact maps were manually edited using the Juicebox Assembly Tools (JBAT) v1.11.08 (RRID:SCR_021172) [40] to produce the expected 17 chromosomes per haplome. Contigs containing assembled telomeres were correctly oriented to the terminal ends by searching for the TTTAGGG repeat (or the reverse complement CCCTAAA) using the analyze_genome function of GENESPACE [43]. Chromosomes were numbered and oriented using haplome A of the 'Gala' assembly [27]. Genome quality and completeness was assessed using benchmarking universal single-copy gene orthologs (BUSCO v5.2.2 (RRID:SCR_015008)) [44] with the "eudicots_odb10" database. Haplome completeness was also assessed using Merqury v1.3 [45].

## Transcriptome sequencing

To facilitate gene annotation, total RNA was isolated from various tissues harvested from 'Honeycrisp', 'Red Delicious', and 'Granny Smith' apple trees grown at the WSU Sunrise Research Orchard near Rock Island, WA USA; 'Gala' and 'WA38' apple trees grown at the WSU and USDA-ARS Columbia View Research Orchard near Orondo, WA USA; and 'D'Anjou' pear trees grown at the WSU Tree Fruit Research and Extension Center Research Orchard in Wenatchee, WA USA using a modified CTAB/Chloroform extraction [46]. Total RNA was assessed for quality (RNA integrity number (RIN) ≥ 8) and purity (A260/280 > 1.8). Sources for all RNA are available in Table 1. Total RNA (2 μg) was used to construct Illumina TruSeq stranded libraries following manufacturers' instructions. Libraries were sequenced on an Illumina NovaSeq 6000 with PE150 reads at the HudsonAlpha Institute for Biotechnology in Huntsville, AL USA.

## Repeat analysis and gene annotation

Repetitive elements on both haplotypes were annotated using EDTA v2.0.0 [47] with flags "–genome, –anno 1, –sensitive=1". To supplement *ab initio* gene predictions, extensive

**Table 1.** Yield of Illumina transcriptome sequencing of fruit, leaves, and flower tissues of apples and pear generated and used for genome annotation in this study.

| Cultivar | Tissue | Reads | Yield (Gbp) | Yield P20 (Gbp) | Average read length | NCBI SRA |
|---|---|---|---|---|---|---|
| Honeycrisp | Fruitlet stage 1 | 45,773,784 | 13,823,682,768 | 13,069,822,280 | 142 | SAMN29611971 |
| | Fruitlet stage 2 | 35,618,706 | 10,756,849,212 | 10,227,275,771 | 143 | SAMN29611972 |
| | Budding leaves | 81,448,971 | 24,597,589,242 | 22,769,634,770 | 139 | SAMN29611973 |
| | Expanding leaves | 35,381,039 | 10,685,073,778 | 9,971,308,535 | 141 | SAMN29611974 |
| | Half-inch terminal buds | 47,811,924 | 14,439,201,048 | 13,409,542,519 | 140 | SAMN29611975 |
| | Flower buds | 45,822,773 | 13,838,477,446 | 13,175,876,315 | 144 | SAMN29611976 |
| | Open flowers | 30,938,395 | 9,343,395,290 | 8,718,474,885 | 141 | SAMN29611977 |
| Gala | Fruitlet stage 1 | 80,440,219 | 24,292,946,138 | 22,928,129,883 | 142 | SAMN29611954 |
| | Fruitlet stage 2 | 32,475,136 | 9,807,491,072 | 9,284,944,973 | 143 | SAMN29611955 |
| | Budding leaves | 30,368,057 | 9,171,153,214 | 8,508,033,713 | 140 | SAMN29611956 |
| | Expanding leaves | 40,650,277 | 12,276,383,654 | 11,306,267,120 | 138 | SAMN29611957 |
| | Roots from tissue culture | 35,324,786 | 10,668,085,372 | 9,940,132,737 | 140 | SAMN29611958 |
| | Quarter-inch terminal buds | 37,532,631 | 11,334,854,562 | 10,634,379,784 | 141 | SAMN29611959 |
| | Flower buds | 39,636,821 | 11,970,319,942 | 11,141,652,382 | 140 | SAMN29611960 |
| | Open flowers | 34,363,075 | 10,377,648,650 | 9,775,838,818 | 142 | SAMN29611961 |
| Red Delicious | Fruitlet stage 2 | 27,319,955 | 8,250,626,410 | 7,682,200,349 | 140 | SAMN29611962 |
| Granny Smith | Fruitlet stage 1 | 29,426,606 | 8,886,835,012 | 8,335,731,187 | 141 | SAMN29611963 |
| | Fruitlet stage 2 | 72,205,133 | 21,805,950,166 | 20,663,261,900 | 143 | SAMN29611964 |
| | Budding leaves | 57,244,195 | 17,287,746,890 | 16,179,280,911 | 141 | SAMN29611965 |
| | Expanding leaves | 40,798,422 | 12,321,123,444 | 11,499,303,808 | 140 | SAMN29611966 |
| | Roots from tissue culture | 32,493,822 | 9,813,134,244 | 9,207,784,729 | 141 | SAMN29611967 |
| | Quarter-inch terminal buds | 30,394,263 | 9,179,067,426 | 8,512,945,196 | 140 | SAMN29611968 |
| | Flower buds | 29,735,514 | 8,980,125,228 | 8,364,532,017 | 140 | SAMN29611969 |
| | Open flowers | 34,303,317 | 10,359,601,734 | 9,603,420,430 | 140 | SAMN29611970 |
| WA 38 | Fruitlet stage 1 | 45,284,208 | 13,675,830,816 | 12,831,991,620 | 141 | SAMN29611978 |
| | Fruitlet stage 2 | 25,486,256 | 7,696,849,312 | 7,261,195,330 | 142 | SAMN29611979 |
| | Budding leaves | 39,339,589 | 11,880,555,878 | 11,017,185,994 | 140 | SAMN29611980 |
| | Expanding leaves | 34,784,980 | 10,505,063,960 | 9,719,694,010 | 139 | SAMN29611981 |
| | Roots from tissue culture | 33,935,508 | 10,248,523,416 | 9,426,506,860 | 138 | SAMN29611982 |
| | Quarter-inch terminal buds | 88,677,165 | 26,780,503,830 | 24,913,194,030 | 140 | SAMN29611983 |
| | Flower buds | 23,170,354 | 6,997,446,908 | 6,588,921,074 | 142 | SAMN29611984 |
| | Open flowers | 35,274,250 | 10,652,823,500 | 9,941,466,644 | 141 | SAMN29611985 |
| D'Anjou | Fruitlet stage 1 | 89,462,306 | 27,017,616,412 | 25,459,693,894 | 142 | SAMN29611986 |
| | Fruitlet stage 2 | 48,481,031 | 14,641,271,362 | 13,921,844,851 | 143 | SAMN29611987 |
| | Budding leaves | 29,823,484 | 9,006,692,168 | 8,442,259,663 | 141 | SAMN29611988 |
| | Expanding leaves | 57,920,009 | 17,491,842,718 | 16,460,531,509 | 142 | SAMN29611989 |
| | Quarter-inch terminal buds | 40,966,825 | 12,371,981,150 | 11,476,090,088 | 140 | SAMN29611990 |
| | Flower buds | 29,183,231 | 8,813,335,762 | 8,264,473,671 | 141 | SAMN29611991 |
| | Open flowers | 32,128,369 | 9,702,767,438 | 8,996,878,963 | 140 | SAMN29611992 |

Gbp: gigabase pairs; NCBI: National Center for Biotechnology Information; SRA: Sequence Read Archive.

extrinsic gene annotation homology evidence is needed. Thus, we downloaded existing RNA-seq data for 'Honeycrisp' apples from the National Center for Biotechnology Information (NCBI) using Sequence Read Archive (SRA) toolkit v2.9.6-1 (SRX3408575, SRX5369275, SRX5369276, SRX5369290, SRX5369299, SRX5369300, SRX5369302, SRX8712695 and SRX8712718) [48–50], and combined with the RNA-seq data generated for this project (described above). We *de novo* assembled these two sets of RNA transcripts separately using Trinity v2.13.2 (RRID:SCR_013048) [51], where we used the flag–trimmomatic to filter the reads for quality. Because the newly generated RNA-seq data were strand-specific, for these we also used the flag "–SS_lib_type RF". We identified open reading frames using TransDecoder v5.5.0 (RRID:SCR_017647) [52]. Gene annotation was performed using BRAKER2 v2.1.6 (RRID:SCR_018964) [53], where we ran BRAKER2 twice, with RNA-seq data

and protein databases run separately. For the RNA-seq run, we first filtered the data for adapters and quality using TRIMMOMATIC v0.39 (RRID:SCR_011848) [54] with leading and trailing values of 3, sliding window of 30, jump of 10, and a minimum remaining read length of 40. We next mapped these data to the genome using STAR v2.7.9a [55] and combined the BAM files using SAMtools (RRID:SCR_005227) [56]. For the homology-based annotation in BRAKER2, we used gene models from *Malus domestica* 'Gala' diploid v2, *M. sieversii* diploid v2 [27], *M. baccata* v1 [57]. *M. domestica* 'Golden Delicious' double haploid v1 (GDDH13) [31], *Pyrus communis* 'Barlett' double haploid v2 [58], and our *de novo* assemblies, in addition to the viridiplantae OrthoDB (RRID:SCR_011980) [59]. We filtered the resulting AUGUSTUS [53] output for those that contained full hints (gene model support) and combined the two runs using TSEBRA v1.0.3 [60]. Finally, we removed any transcript/gene with ≥90% softmasking, i.e., mainly repeat sequences. Genome annotation completeness of our genome and other *Malus* genomes were assessed using BUSCO v5.2.2 (RRID:SCR_015008) [44] with the "eudicots_odb10" database for comparative purposes.

The final 'Honeycrisp' gene sets from both haplomes were annotated with InterProScan v5.44–79.0 (RRID:SCR_005829) [61, 62], including a search against all the available InterPro databases and Gene Ontology (GO) [63, 64] prediction. In addition, genes were searched against the 26Gv2.0 OrthoFinder v1.1.5 (RRID:SCR_017118) [65] gene family database using both BLASTp (RRID:SCR_001010) [66] and HMMscan (RRID:SCR_00530) 5 [67] classification methods with the GeneFamilyClassifier tool from PlantTribes 2 [68]. This analysis provided additional functional annotation information that includes gene counts of scaffold taxa, superclusters at multiple clustering stringencies, and functional annotations that were pulled from various public genomic databases.

## COMPARATIVE GENOMICS

Similarities in lengths and structural variations between the two haplomes were determined by running MUMmer v4.0 (RRID:SCR_018171) [69] and Assemblytics [70]. To identify the shared and unique gene families among *Malus* species and cultivars, genes from the six publicly available *Malus* genomes (Table 2) were integrated into the PlantTribes 2 gene model database (26Gv2.0) using the same method described above. The overlapping orthogroups (with at least 30 counts in the category) among the eight *Malus* annotations (including both haplomes from 'Honeycrisp') were calculated and visualized with an upset plot generated by TBtools v1.0986982 [71].

## DATA VALIDATION AND QUALITY CONTROL

### A haplotype-phased chromosome-scale assembly

In total, nearly 55× coverage of PacBio HiFi reads and nearly 200× coverage of Dovetail Omni-C reads (Table 3) was generated. This included 2,543,518 HiFi reads with an average length of 14,655 base pairs (bp) and ~91% of reads ≥10,000 bp. Two phased haplomes, haplome A (HAP1) and haplome B (HAP2, these two sets of terms will henceforth be used interchangeably), were assembled and validated by inspection of the Omni-C contact maps (Figure 2). Both haplomes are highly contiguous and of similar size. HAP1 is 674 megabase pairs (Mbp) in length, contained in 473 contigs with a contig $N_{50}$ of 32.8 Mbp, whereas HAP2 is 660 Mbp in length, contained in 215 contigs with a contig $N_{50}$ of 31.6 Mbp (Table 4). No mis-joins requiring manual breaks were identified in the assemblies. For HAP1, a total of 13



**Table 2.** Comparison of genomic features and assembly statistics of current assembly of 'Honeycrisp' genome and previously published genomes of apples.

| Genomes | 'Honeycrisp' (reference: this work) | 'Gala', *M. sieversii*, *M. sylvestris* (all Diploid) [72] | HFTH1; 'Hanfu' (Triploid) [28] | GDDH13; 'Golden Delicious' (Double haploid) [29] | 'Golden Delicious' (Diploid) [24] |
|---|---|---|---|---|---|
| **Assembly** | | | | | |
| Haploid genome size (Mbp) | 660–674 | 666–679 | 658.9 | 651 | 742 |
| Scaffold N50 (Kbp) | 31.6–32.8 | 6.1–21.8 | 6.99 | 5.5 | 16 |
| Complete BUSCO (%) | 98.6–98.7 | 98.0–98.8 | 98.6 | 98.0 | 82.0 |
| **Annotation** | | | | | |
| Protein-coding genes | 47,563–48,655 | 44,691–44,847 | 44,677 | 42,140 | 57,386 |
| Complete BUSCO (%) | 96.8–97.4 | 94.6–95.4 | 93.6 | 96.1 | 68.0 |
| **Gene family** | | | | | |
| Number of orthogroups in 26Gv2 | 10,351–10,367 | 10,044–10,115 | 9974 | 10,117 | 8824 |

BUSCO: Benchmarking Universal Single-Copy Orthologs; Kbp: kilobase pairs; Mbp: megabase pairs.

**Table 3.** Overview of PacBio HiFi and Omni-C sequencing data generated for the 'Honeycrisp' genome assembly.

| Library | Sequencing | Length (Nucleotides) | Number of reads |
|---|---|---|---|
| JNQN | Omni-C | 150 | 951,241,272 |
| HiFi-1 | PacBio HiFi | 14,881* | 1,088,992 |
| HiFi-2 | PacBio HiFi | 14,429* | 1,454,526 |

* Average length.

**Table 4.** Summary of 'Honeycrisp' genome assembly statistics.

| Assembly | Length | # Contigs | Longest contig (bp) | N50 | L50 | QV | *k*-mer completeness (%) | BUSCO (%) |
|---|---|---|---|---|---|---|---|---|
| Honeycrisp Haplome A | 674,476,353 | 473 | 55,653,390 | 32,818,622 | 9 | 64.5 | 82.7 | 98.6 |
| Honeycrisp Haplome B | 660,238,068 | 215 | 56,154,892 | 31,578,807 | 9 | 66.7 | 83 | 98.7 |
| Combined | | | | | | 65.5 | 98.6 | |

QV: quality value.

joins were made to build the final assembly into 17 chromosomes, with 95.4% of the assembled sequence contained in the 17 pseudomolecules representing chromosomes. Nineteen joins were made for HAP2, with 98.2% of the assembled sequence in the 17 pseudomolecules. Based on the Merqury *k*-mer analysis (Figure 3), the HAP1 assembly had a *k*-mer completeness of 82.7% (quality value [QV] 64.5), the HAP2 assembly 83% (QV 66.7), and the combined assemblies were 98.6% (QV 65.5) (Table 4). BUSCO completeness of HAP1 was 98.6% and HAP2 98.7%, suggesting high genome completeness for both haplomes, comparable or superior to other high quality apple genome assemblies (Table 2). The two haplomes are structurally similar to each other (Figure 4). Compared with the assembly statistics of previously published apple genomes, the current 'Honeycrisp' assemblies are the most contiguous to date (Table 2).

## Genome annotation

The yield of Illumina transcriptome sequencing data of fruit, leaves, and flower tissues of apples and pear ranged from approximately 9 to 27 gigabase pairs (Gbp) in flowers and leaf buds respectively (Table 1). Nearly 62% of both haplomes were annotated as repetitive DNA, mostly comprised of long terminal repeat (LTR) retrotransposons (Table 5). A total of 47,563 genes were annotated in HAP1 and 48,655 in HAP2, slightly more than in other published *Malus* annotations (Table 2). Complete BUSCO scores of the protein annotations are 96.8%

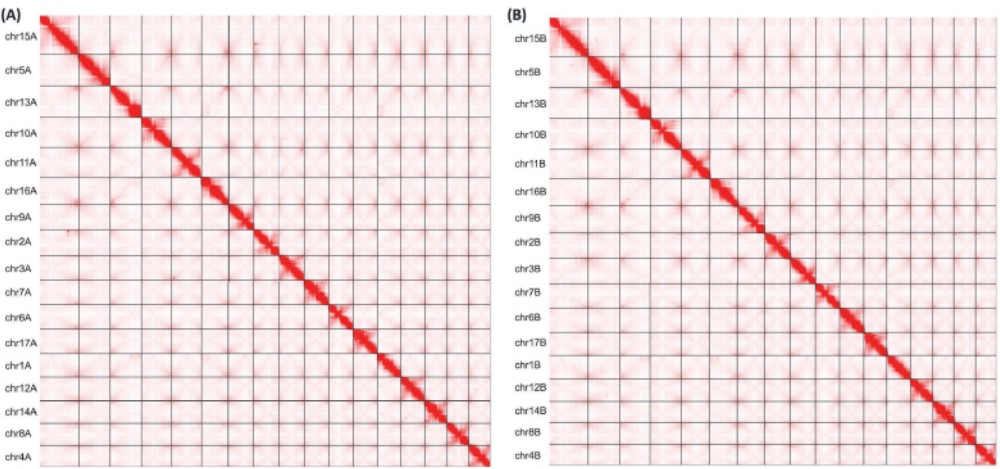

**Figure 2.** **Omni-C contact maps of the assembled chromosome-length scaffolds of 17 chromosomes.**
(A) Haplome A and (B) Haplome B of 'Honeycrisp' genome.

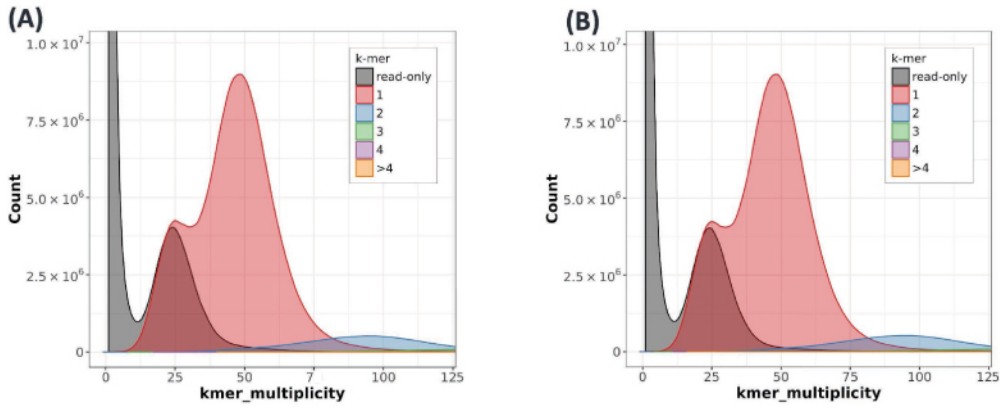

**Figure 3.** **Histogram of *k*-mer multiplicity of sequence reads.**
(A) Haplome A and (B) Haplome B of 'Honeycrisp' genome assemblies. *k*-mer multiplicity (*x*-axis) is plotted against *k*-mer counts (*y*-axis) to estimate the heterozygosity, copy numbers, sequencing depth, and completeness of a genome using Merqury v1.3 [45]. Colors in the plot represent the number of times each *k*-mer is found in the genome assembly.

for HAP1 and 97.4% for HAP2, the highest completeness among all publicly available *Malus* genome annotations (Table 2). 72.85% and 68.88% of the predicted transcripts were annotated with Interpro terms, 68.58% and 64.94% with Pfam domains, and 51.04% and 48.76% with at least one GO term in HAP1 and HAP2, respectively. In the PlantTribes 2 classification, 91.11% and 85.50% of the predicted transcripts from HAP1 and HAP2, respectively, were assigned to pre-computed orthogroups.

As the number of plant genomes are being generated at an unprecedented speed, we developed the following gene naming convention to avoid potential ambiguity:

Maldo.hc.v1a1.ch10A.g00001.t1 – where: Maldo means *Malus domestica*; hc is the cultivar, 'Honeycrisp'; v1a1 indicates the first assembly and first annotation of this genome; ch10A identifies the gene as annotated from chromosome 10 (versus from an unplaced scaffold, which will be indicated by "sc") in haplome A (HAP1) (versus haplome B (HAP2));

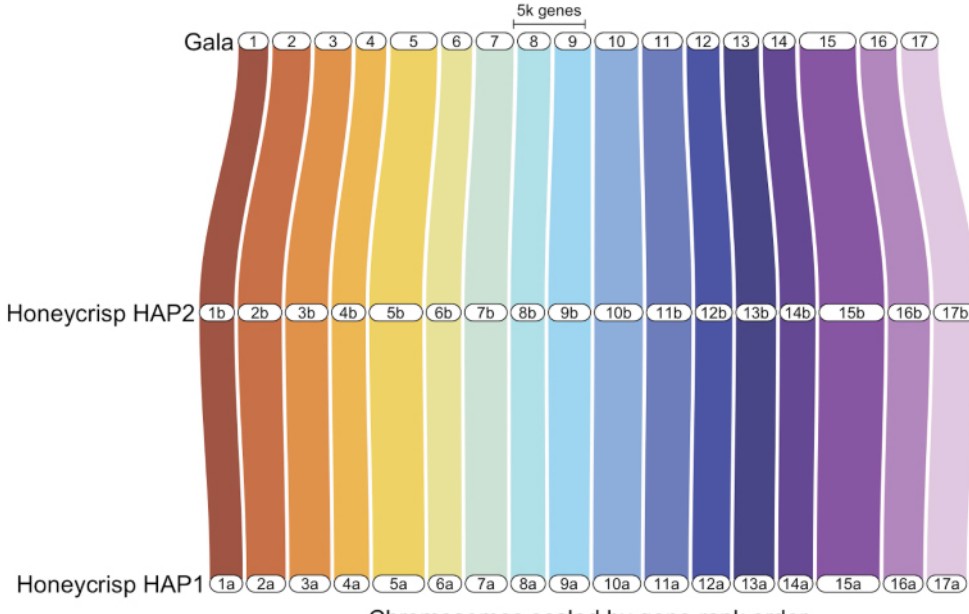

**Figure 4.** Synteny comparison of 'Honeycrisp' Haplome 1 (HAP1), 'Honeycrisp' Haplome 2 (HAP2) from this study, and 'Gala' [27] genomes.
GENESPACE [43] was used for synteny comparison.

**Table 5.** Summary of repetitive element annotation in Haplome A and Haplome B of the 'Honeycrisp' genome assemblies.

| Class | | Haplome A (%) | Haplome B (%) |
|---|---|---|---|
| LTR | | | |
| | Copia | 9.73 | 9.60 |
| | Ty3 | 20.29 | 17.80 |
| | unknown | 14.89 | 16.86 |
| TIR | | | |
| | CACTA | 2.21 | 1.95 |
| | Mutator | 4.16 | 4.25 |
| | PIF Harbinger | 2.43 | 2.60 |
| | Tc1_Mariner | 0.15 | 0.27 |
| | hAT | 2.30 | 2.31 |
| | polinton | – | 0.01 |
| nonLTR | | | |
| | LINE_element | 0.18 | 0.17 |
| | unknown | 0.09 | 0.18 |
| nonTIR | | | |
| | helitron | 2.95 | 3.18 |
| repeat region | | 2.91 | 2.78 |
| **Total** | | **62.43** | **61.97** |

g00001 is a five-digit gene identifier; and t1 represents a transcript number of the gene.

### Gene family analysis

Gene family evaluation was performed using PlantTribes 2 and its 26Gv2-scaffold orthogroup database, which contains representative protein coding sequences from most

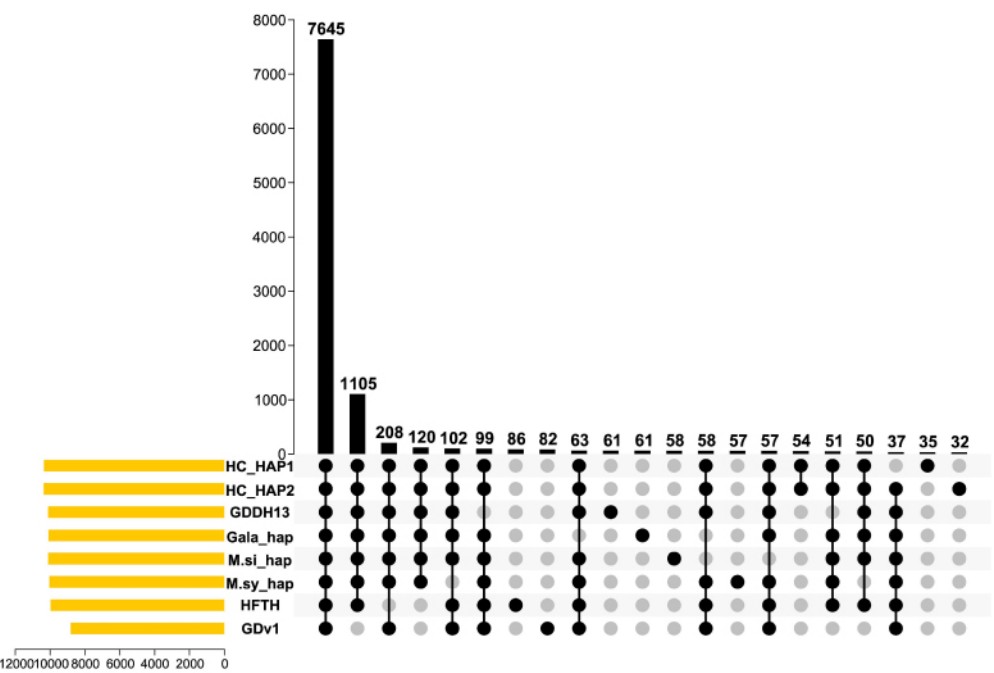

**Figure 5.** The Honeycrisp genome captured a vast majority of *Malus* gene families.
Black dots indicate presence of gene families and gray dots indicate absence. Yellow horizontal bars represent the number of orthogroups in each genome. The black vertical bars represent the number of orthogroups in each category. HC: 'Honeycrisp' (this work); GDDH13: *Malus domestica* GDDH13; Gala_hap: *M. domestica* 'Gala' haploid; M.si_hap: *M. sieversii* haploid; M.sy_hap: *M. sylvestris* haploid; HFTH: *M. domestica* HFTH1; GDv1: *M. domestica* Golden Delicious v1.

major land plant lineages. A total of 11,263 unique orthogroups (OGs) were identified in all eight *Malus* annotations (including the two 'Honeycrisp' haplomes) investigated. 'Honeycrisp' transcripts were assigned to 10,351 and 10,367 orthogroups, similar to 'Gala' and GDDH13 (Table 2 and Figure 5). We further investigated orthogroups that are shared and unique in the eight *Malus* annotations. Most (7645) orthogroups are shared by all the genomes, and 9279 orthogroups were shared by both 'Honeycrisp' haplomes and five other genomes (Figure 5). This comparison indicates that the 'Honeycrisp' annotation captured genes in virtually all the *Malus* gene families. We also found 54 orthogroups unique to 'Honeycrisp' (i.e., shared by the two 'Honeycrisp' haplomes only) and 35 and 32 that are unique to each 'Honeycrisp' haplome (Figure 5). These orthogroups could provide valuable information in the molecular mechanisms underlying genotype-specific traits.

## RE-USE POTENTIAL

This fully phased, high-quality, chromosome-scale genome of 'Honeycrisp' apple will add to the toolbox for apple genetic research and breeding. It will enable genetic mapping, identification of genes, and development of molecular markers linked to disease, pest resistance, abiotic stress tolerance and adaptation, as well as horticulturally relevant harvest and postharvest fruit quality traits for use in apple breeding programs. Ultimately, the addition of high-quality genomic resources for 'Honeycrisp' can lead to enhanced orchard and supply chain management for many other apple cultivars, promoting future sustainability of the pome fruit industry.

## DATA AVAILABILITY

The whole genome sequence data generated in this study have been deposited at the NCBI database under BioProject ID PRJNA791346. PacBio HiFi reads, and Hi-C reads are deposited in NCBI with the SRA accession number SAMN24287034 and SAMN29611953, respectively. Transcriptomic data generated in this study for genome annotation are deposited in NCBI with SRA accession numbers from SAMN29611954 to SAMN29611992. The Maldo.hc.v1a1 'Honeycrisp' genome assembly, gene annotation, and functional annotation for both haplomes can be accessed *via* the *GigaScience* GigaDB repository [73], and will be available in the Genomic Database for Rosaceae, which is currently in progress.

## DECLARATIONS
## LIST OF ABBREVIATIONS

BLAST: Basic Local Alignment Search Tool; bp: Base Pair; BUSCO: Benchmarking Universal Single Copy Orthologs; Gb: gigabases; GO: Gene Ontology; HMW: High-molecular-weight; JBAT: Juicebox Assembly Tools; LTR: Long Terminal Repeat; NCBI: National Centre for Biotechnology Information; OG: Orthogroup; QV: Quality Value; RIN: RNA Integrity Number; SMRT: Single Molecule Real Time; TRF: Tandem Repeat Finder

## ETHICAL APPROVAL

Not applicable.

## CONSENT FOR PUBLICATION

Not applicable.

## COMPETING INTERESTS

The authors declare that they have no competing interests.

## FUNDING

This research was funded by Washington Tree Fruit Research Commission (grant number: AP-19-103), LH; United States Department of Agriculture, Agricultural Research Service, LH; New York State Department of Agriculture & Markets, Apple Research & Development Program (ARDP), grant number: CM04068AQ, AK.

## AUTHORS' CONTRIBUTION

AK, and LH conceptualized, designed, and managed the project. SBC, HZ, AH, and HH, constructed DNA and RNA, and RNA-seq libraries for sequencing. SBC, and HZ, performed genome and transcriptome sequence analysis and interpretation. SBC, HZ, AS, HH, AH, LH, and AK drafted, revised, and finalized the manuscript. All authors read and approved the final version.

## ACKNOWLEDGEMENTS

We acknowledge Della Cobb-Smith and Jugpreet Singh for their help in sample collection and preparing genomic DNA extraction.

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
