## [Reviewer Report]

Reviewer name and names of any other individual's who aided in reviewer zhang liyi Do you understand and agree to our policy of having open and named reviews, and having your review included with the published papers. (If no, please inform the editor that you cannot review this manuscript.)YesIs the language of sufficient quality?YesPlease add additional comments on language quality to clarify if needed
Are all data available and do they match the descriptions in the paper? YesAdditional CommentsAre the data and metadata consistent with relevant minimum information or reporting standards? See GigaDB checklists for examples <a href="http://gigadb.org/site/guide" target="_blank">http://gigadb.org/site/guide</a>YesAdditional CommentsIs the data acquisition clear, complete and methodologically sound?YesAdditional CommentsIs there sufficient detail in the methods and data-processing steps to allow reproduction?YesAdditional CommentsIs there sufficient data validation and statistical analyses of data quality? YesAdditional CommentsIs the validation suitable for this type of data?YesAdditional CommentsIs there sufficient information for others to reuse this dataset or integrate it with other data?YesAdditional CommentsAny Additional Overall Comments to the AuthorRecommendationAccept

---

## [Reviewer Report]

Reviewer name and names of any other individual's who aided in reviewer Luca BiancoDo you understand and agree to our policy of having open and named reviews, and having your review included with the published papers. (If no, please inform the editor that you cannot review this manuscript.)YesIs the language of sufficient quality?YesPlease add additional comments on language quality to clarify if needed
Are all data available and do they match the descriptions in the paper? NoAdditional CommentsI could not access to the Bioproject data nor see the results files (i.e. fasta, gff,...) but I am confident they will be available once the paper is acceptedAre the data and metadata consistent with relevant minimum information or reporting standards? See GigaDB checklists for examples <a href="http://gigadb.org/site/guide" target="_blank">http://gigadb.org/site/guide</a>YesAdditional CommentsIs the data acquisition clear, complete and methodologically sound?YesAdditional CommentsIs there sufficient detail in the methods and data-processing steps to allow reproduction?YesAdditional CommentsIs there sufficient data validation and statistical analyses of data quality? YesAdditional CommentsThe only exception is what I mentioned regarding the haplotype separation (see general comments below)Is the validation suitable for this type of data?YesAdditional CommentsIs there sufficient information for others to reuse this dataset or integrate it with other data?YesAdditional CommentsThis paper describes the genome sequence of Honeycrisp, an important apple cultivar, produced with the latest sequencing technologies and assembled into phased chromosomes. In my opinion, the manuscript is well written, very interesting and certainly worth publication. There are only a few points that I would like to see addressed: 

1) How can you be sure that the two haplomes are a good representation of each chromosome and not a mix of the two haplotypes? In other words, have you checked that the whole sequence of each chromosome represents one phase only? It would be great if you could provide some data (e.g. SNPs,...)  to support this and discuss the results obtained in this regard. 

2) Some additional stats regarding the obtained sequence could be added to table 2 and/or table 5 (e.g. number of Ns in the genome, how many telomers were assembled in each chromosome -- if not all telomers were identified, )

3) The gene family analysis among the different apple genomes is quite interesting but rather superficial. It would be nice to dig deeper into the function of the orthogroups that are unique to Honeycrisp, describe what pathways they are involved in and so on... Any Additional Overall Comments to the AuthorRecommendationMinor Revision